# Diagnosis of *Taenia solium* infections based on "mail order" RNA-sequencing of single tapeworm egg isolates from stool samples

**Henrik Sadlowski**[1], **Veronika Schmidt**[2,3], **Jonathan Hiss**[1], **Johannes A. Kuehn**[1],
**Christian G. Schneider**[1], **Gideon Zulu**[4], **Alex Hachangu**[4], **Chummy S. Sikasunge**[4],
**Kabemba E. Mwape**[4], **Andrea S. Winkler**[2,3], **Markus Schuelke**[1,5]*

**1** Charité-Universitätsmedizin Berlin, corporate member of the Freie Universität Berlin and Humboldt-Universität zu Berlin, NeuroCure Cluster of Excellence, Berlin, Germany, **2** Department of Neurology, Centre for Global Health, Klinikum rechts der Isar, Technical University Munich (TUM), Munich, Germany, **3** Centre for Global Health, Institute of Health and Society, University of Oslo, Oslo, Norway, **4** School of Veterinary Medicine, University of Zambia, Lusaka, Zambia, **5** Charité-Universitätsmedizin Berlin, corporate member of the Freie Universität Berlin and Humboldt-Universität zu Berlin, Department of Neuropediatrics, Berlin, Germany

* markus.schuelke@charite.de

**Data Availability Statement:** The raw FASTQ files from the *T. solium* genome sequencing and the *.bam file from the MW718881 mtDNA alignment

## Abstract

Combined community health programs aiming at health education, preventive anti-parasitic chemotherapy, and vaccination of pigs have proven their potential to regionally reduce and even eliminate *Taenia solium* infections that are associated with a high risk of neurological disease through ingestion of *T. solium* eggs. Yet it remains challenging to target *T. solium* endemic regions precisely or to make exact diagnoses in individual patients. One major reason is that the widely available stool microscopy may identify *Taenia* ssp. eggs in stool samples as such, but fails to distinguish between invasive (*T. solium*) and less invasive *Taenia* (*T. saginata*, *T. asiatica*, and *T. hydatigena*) species. The identification of *Taenia* ssp. eggs in routine stool samples often prompts a time-consuming and frequently unsuccessful epidemiologic workup in remote villages far away from a diagnostic laboratory. Here we present "mail order" single egg RNA-sequencing, a new method allowing the identification of the exact *Taenia* ssp. based on a few eggs found in routine diagnostic stool samples. We provide first *T. solium* transcriptome data, which show extremely high mitochondrial DNA (mtDNA) transcript counts that can be used for subspecies classification. "Mail order" RNA-sequencing can be administered by health personnel equipped with basic laboratory tools such as a microscope, a Bunsen burner, and access to an international post office for shipment of samples to a next generation sequencing facility. Our suggested workflow combines traditional stool microscopy, RNA-extraction from single *Taenia* eggs with mitochondrial RNA-sequencing, followed by bioinformatic processing with a basic laptop computer. The workflow could help to better target preventive healthcare measures and improve diagnostic specificity in individual patients based on incidental findings of *Taenia* ssp. eggs in diagnostic laboratories with limited resources.

can be accessed through the GEO database under the accession number GSE175668 (https://www.ncbi.nlm.nih.gov/geo/query/acc.cgi?acc=GSE175668). The sequence describing the T. solium mtDNA is available at GenBank (https://www.ncbi.nlm.nih.gov/nuccore/2085340607). An extended protocol with all detailed steps, instructions, and illustrations can be found on protocols.io: http://dx.doi.org/10.17504/protocols.io.bzx6p7re. We also provide a video demonstrating how to prepare the glass needles from ordinary Pasteur pipettes that are needed to break and open Taenia spp. eggs for single egg RNA-sequencing. The video can be accessed through http://dx.doi.org/10.6084/m9.figshare.16955734.

**Funding:** This study has been funded by the Validation fund of the Berlin Institute of Health to HS and MS, the Deutsche Forschungsgemeinschaft (DFG; German Research Foundation, https://www.dfg.de/) under Germany's Excellence Strategy – EXC-2049-390688087 to MS, and the German Federal Ministry of Education and Research (BMBF, https://www.bmbf.de/) under Research Networks for Health Innovation in Sub-Saharan Africa – CYSTINET-Africa 01KA1618 to ASW. The funders had no role in study design, data collection and analysis, decision to publish, or preparation of the manuscript.

**Competing interests:** The authors have declared that no competing interests exist.

## Author summary

Taeniasis is an infection of the intestine with tapeworms such as *Taenia saginata* (beef tapeworm), *Taenia solium* (pork tapeworm), or *Taenia asiatica* (Asian tapeworm). In the case of pork tapeworm in particular, infected individuals excrete thousands of eggs in their feces, which can cause cysticercosis if ingested by mouth and develop into larvae. Infection can occur through contaminated vegetables, but also by unwashed hands that had contact with *Taenia* eggs, including auto-infection. The larval cysts are deposited throughout the body, including the brain (neurocysticercosis), which is a common cause of epilepsy worldwide. Stool microscopy is the standard method for detecting taeniasis, but it cannot distinguish between infection with the beef and the more dangerous pork tapeworm. We have now developed a next-generation sequencing method that allows us to determine the genetic code of the tapeworm's mitochondrial DNA from just two eggs in a stool sample. The sequence obtained makes it possible to uniquely identify the *Taenia* subspecies, treat patients appropriately, and inform public health efforts of endemic regions in order to eradicate this infection, as proposed by WHO.

## Introduction

The parasite *Taenia solium*, commonly known as the pork tapeworm, is endemic in 57 countries and suspected to be endemic in a further 19 countries and accounts for a total of over 2.7 million Disability Adjusted Life Years (DALY) [1]. The intestinal infestation with tapeworms of the species *T. solium* (pork tapeworm), *T. saginata* (beef tapeworm), and *T. asiatica* ("Asian" tapeworm) is called "taeniasis" and mostly affects people in rural areas, and is classified by the WHO as a "Neglected Tropical Disease" [2]. This term refers to diseases that affect large numbers of people, but are nevertheless neglected in terms of research, diagnostic capabilities, and access to pharmacotherapy.

*Taenia* ssp. carriers are often asymptomatic or show only mild nonspecific symptoms such as abdominal pain and nausea and may be infested with the adult worm for years. Throughout this time, individuals excrete in their stools millions of microscopic *Taenia* ssp. eggs that may survive for up to one year under field conditions [3]. Specifically *T. solium* eggs, if orally ingested, may cause extra-intestinal infections called "cysticercosis" [4]. This may happen *via* autoinfection, by inoculation through other members of the household or community *via* contaminated hands, or through vegetables or fruits. Cysticercosis can be found in different tissues such as muscle, skin, and the spinal cord and/or brain (neurocysticercosis, NCC), with the latter being the most common cause of acquired epilepsy in countries where *T. solium* (neuro) cysticercosis/taeniasis (TSCT) is endemic [5,6]. Taeniasis was present in 16% of NCC patients with mild to moderate infections and in 82% with massive infections, clearly hinting towards continuous autoinfection as a risk factor to develop NCC [7].

Different approaches exist to reduce TSCT disease burden, eventually aiming at elimination of *T. solium* [8,9]. These measures comprise preventive chemotherapy of *T. solium* taeniasis in form of mass drug administration to whole communities comprising humans and pigs [10], community health education, meat inspection in abattoirs as well as mass screenings, and vaccination of pigs. Targeted screening of individuals with epilepsy or with progressively worsening severe chronic headaches may also help to identify and treat NCC [6,11]. However, for lack of expensive laboratory equipment and state sponsored screening programs, application of these measures remains mostly limited to epidemiological research projects while *T. solium* infections are of general health concern, stretching across vast rural areas on multiple

continents. A simple and easy to administer diagnostic test would empower other stakeholders outside of professional research networks to collect regional data on *T. solium* transmission and to advocate for local disease control programs.

In the absence of a laboratory infrastructure, stool microscopy is the only globally available technique allowing the diagnosis of a *Taenia ssp*. infection [12]. Other diagnostic tests, such as nested PCR-based assays [13] or the detection of *T. solium* corpro-antigen by ELISA [14], are often not available outside reference and research laboratories [15]. However, routine stool samples sent to central diagnostic laboratories from remote locations for various reasons other than a suspected taeniasis (e.g. for gastrointestinal problems or diarrhea) could serve as a starting point for epidemiological prevalence screening or public health surveys [16]. In such stool samples, tapeworm eggs may be identified as such, but egg morphology does not allow differentiation between *T. solium* and other *Taenia* ssp. In order to make a definitive diagnosis, e.g. for *T. solium* prevalence screenings, a large effort has to be made by community health workers to **(i)** travel to and (re)locate the patient in a remote village, **(ii)** treat the patient with an antiparasitic drug and purgatives, **(iii)** sieve through the collected stool samples over 1–2 days for obtaining characteristic parasite fragments, such as proglottids or the scolex (head) of the tapeworm. Based on the histological investigation of these fragments from patient stool samples after haematoxylin-eosin staining, an experienced pathologist might come to the correct diagnosis based on fine morphological characteristics of the shape of the uterine cavities in the proglottids [17] or on the characteristic shape of the scolex [18]. These interlinked steps often fail for logistic reasons or due to non-compliance of the patients due to social stigmatization, thereby missing the infestation of a local population with *T. solium*. The problem may also be aggravated by migrant workers, who have to leave their village for extended periods of time to find work far away and cannot be contacted by mobile phone. Hence, a definitive diagnosis of *T. solium* taeniasis is almost never secured in practice, because the required tapeworm fragments are not always excreted and/or of the required quality. This problem is well known, but currently difficult to gauge with findings reaching from zero tapeworm fragments found in 174 treated patients in one study [19] to 42 fragments found in 68 patients who were treated with an optimized protocol in another [20].

Such limitations are a significant roadblock for research into the regional prevalence of *T. solium* and preclude widespread community based identification and treatment of *T. solium* carriers. However, easy to administer tests [21] and accurate prevalence data would be urgently needed for the planning of community health programs at the governmental as well as the WHO-level with the ultimate goal to regionally eliminate *T. solium* [8] by the year 2030 as proposed by the WHO [22].

Here we present for the first time the *T. solium* transcriptome and describe an easy-to-use workflow that harnesses next generation sequencing technology to allow researchers, health organizations, and clinicians to accurately diagnose *T. solium* taeniasis based on the discovery of single *Taenia* ssp. eggs from stool samples.

## Materials and methods

### Ethics statement

All aspects of the study have been approved by the Institutional Review Boards of the University of Zambia, Lusaka, Zambia (approval number IRB 00005948) and the Charité—Universitätsmedizin Berlin, Berlin, Germany (approval number EA4/147/17). The CYSTINET-Africa project received approval from the Ethics Committee of the Klinikum rechts der Isar, Technical University of Munich, Germany (approval number 537/18 S-KK). Written informed consent was obtained from all participants of the study.

## Estimation of the worldwide prevalence of *Taenia solium* infections based on published data

In order to illustrate the importance of *T. solium* surveys and to estimate the worldwide risk for a *T. solium* infection and to highlight regions with a high incidence, we combined data from the World Health Organization (WHO) on the epidemiology of *T. solium* infections with World Bank data on the percentage of people living in rural areas. Based on the WHO data, we created a list of countries where *T. solium* is, or is suspected to be endemic. Based on World Bank data, we then counted the number of people living in these countries in rural areas in order to define a population per country with a higher risk of being exposed to *T. solium* and assigned a color scheme to highlight countries with a large population potentially exposed to *T. solium*. In order to visualize how much original data are reported on *T. solium* infections in the respective countries, we performed a PubMed search at November 2020 using the search terms ("*taenia solium*" NOT review[publication type] AND solium[title/abstract] AND study[title/abstract] AND 2010:2020[date—publication]) and recovered 384 articles. We manually curated this dataset and selected a total of 176 publications that report actual prevalence estimates for pig or human *T. solium* infections and extracted all geographic information provided in title, abstract, and author affiliation from these publications. All locations were then plotted onto a world map to visualize the intensity of research efforts into *T. solium*.

## Sample collection

This study was conducted between December 2018 and August 2020 in collaboration with the Zambian and German study team of the CYSTINET-Africa network (https://www.cystinet-africa.net/) in four villages in the Eastern Province around Chipata town in Zambia. As part of the epidemiological activities of the Zambian study team at the School of Veterinary Medicine, University of Zambia (UNZA), 600 stool samples were collected to be further screened for *Taenia* ssp. eggs using classical stool microscopy. The *Taenia* spp. egg positive samples then further served as source material for the present study. The stool samples were stored at 4°C for up to five days in the field laboratory and then transported over 1–3 days on coolpacks to the UNZA regional reference laboratory that was connected to an international mail service. However, we did not perform a systematic time series to investigate how long the intact eggs could be stored in the field to obtain a good RNA sequencing result. The longest storage time was 8 days, which would be the maximum transport time from remote rural areas to a laboratory with access to international postal services. Considering that eggs in the wild can survive for several weeks or even months and remain infectious [3], i.e. "biologically intact", we assume that much longer storage times would be possible, provided that the eggs remain intact.

## Stool microscopy

Examination for the presence of helminth eggs was performed as reported previously [23] but in native unmodified stool samples. One gram of native stool sample was resuspended in 10 volumes of phosphate buffered saline (PBS) and centrifuged at 2,000 x *g* for 20 minutes at room temperature. After removal of the supernatant, the stool pellet was resuspended in two volumes of PBS. One drop of the resuspended stool matrix was then put on a microscopic glass slide, covered with a cover slip, and inspected with a Leica DMiL phase contrast microscope with a HI Plan 20x lens for the presence of *Taenia* ssp. eggs.

## Preparing reagents and tools

A detailed step-by-step protocol has been uploaded to protocols.io which can be accessed *via* the DOI dx.doi.org/10.17504/protocols.io.bzx6p7re. Upon identification of *Taenia* ssp. eggs, we ordered the reagents for an optimized version of the SMART-Seq buffer (Takara Bio Inc, Kusatsu, Japan, containing the following components: dNTPs, RNasin Ribonuclease Inhibitor, and Triton X-100). After delivery, the dry ice was kept and the material for the preparation of *Taenia* eggs was set up: glass needles for disruption of the robust *Taenia* eggs were manufactured using minimal resources by **(i)** carefully melting borosilicate Pasteur pipettes on a Bunsen burner, **(ii)** pulling them gently to narrow the glass tube and to form a narrowly tapering tip, **(iii)** at the same time twisting both ends against each other in order to seal the lumen at the tip of the needle **(iv)** before breaking it. An illustration of this procedure is provided on **Fig 1** and in a video describing the entire procedure on http://dx.doi.org/10.6084/m9.figshare.16955734. The heating of the glass capillaries rendered the tips RNAse-free. Hence, we suggest performing this procedure only shortly before using the capillaries and prevent skin contact of their tips. Once the glass had cooled, the needles were tested for the absence of a lumen by tipping them into a 5 μL drop of diethyl pyrocarbonate (DEPC)-treated RNAse-free water on a microscopic slide and visually inspecting whether water would be soaked into the needle by capillary forces or not. Only needles that failed to do so were used for subsequent steps.

## Retrieving *Taenia* spp. eggs

In order to set up and evaluate the sequencing protocol, we generated three aliquots (technical replicates) of the *Taenia* ssp. positive stool sample and froze them at -80˚C. After thawing, the

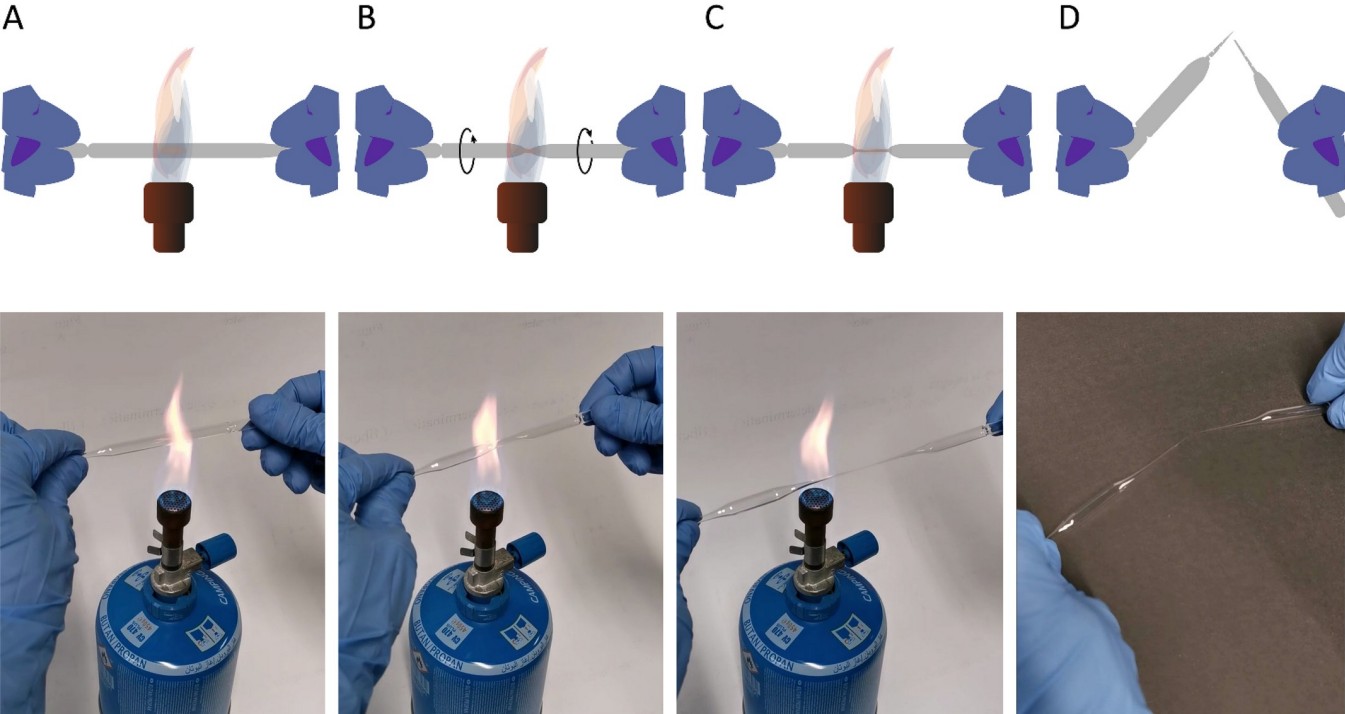

**Fig 1. Preparation of borosilicate needles for egg disruption. (A)** Heat a borosilicate pipette on a Bunsen burner until the glass starts to melt, **(B)** twist the pipette once the glass starts to become viscous, **(C)** pull the ends of the pipet carefully until there remains only a hair-thin connection between the two ends, **(D)** remove the pipet from the flame, let it cool, and finally break the thin connection.

samples (samples #1 and #2, biological replicates) were processed immediately. Sample #3 was kept at 4°C for 48 hours before further processing. We resuspended each sample in 5 volumes of PBS and placed 40 μL of the sample to a glass slide without cover slip. Single *Taenia* eggs were aspirated under the inverted phase contrast microscope with a normal 10 μL Eppendorf laboratory pipette and transferred into a fresh drop of PBS on the same slide. Three successive washing steps were performed by passing the *Taenia* eggs between the PBS droplets in order to remove the adhering stool matrix. We repeated this process twice and created duplicates from one stool sample, each containing two clean *Taenia* eggs. These replicates were then used for the disruption process using borosilicate pipettes, for PCR-confirmation of the presence of *T. solium* DNA, and for RNA-sequencing.

## DNA cleanup for *Taenia solium* specific PCR

8 *Taenia* eggs in total were isolated from the stool sample and placed in 500 μL PBS. DNA was extracted from the eggs using the bead-based Quick-DNA Tissue/Insect Microprep Kit (D6015, Zymo Research, Freiburg, Germany) according to the manufacturer's instructions. 2 μL of the eluate were used for a nested PCR, based on a published protocol [13] amplifying the *T. solium* specific TSO31 gene using the outer primers 5'-ATG ACG GCG GTG CGG AAT TCT G-3' (forward) and 5'-TCG TGT ATT TGT CGT GCG GGT CTA C-3' (reverse) followed by the nested inner primers 5'-GGT GTC CAA CTC ATT ATA CGC TGT G-3' (forward) and 5'-GCA CTA ATG CTA GGC GTC CAG AG-3' (reverse) ultimately amplifying a 234 bp fragment.

## Disruption of the *Taenia* spp. eggs and sample preparation for shipping

The sequencing reactions from the two stool replicates [sample #1 (En1dxA) and #2 (EN2dxA)] were prepared by placing the cleaned *Taenia* eggs in a total volume of 4 μL of PBS on a fresh cover slide and by mounting the slide to a Leica DMiL phase contrast microscope with a HI Plan 20x lens. The two freshly prepared borosilicate needles were inserted into the drop and then located under visual control next to the *Taenia* egg. The two needless were then moved towards each other squeezing and ultimately disrupting the egg as detailed in **Fig 2**. This procedure was repeated for the second egg. The entire drop with the cell lysate was immediately transferred into a 500 μL tube containing 0.6 μL of the optimized SMART-Seq buffer, placed on dry ice and shipped on dry ice by an international courier service to an NGS-sequencing facility (BGI, Shenzhen, Guangdong, PR China).

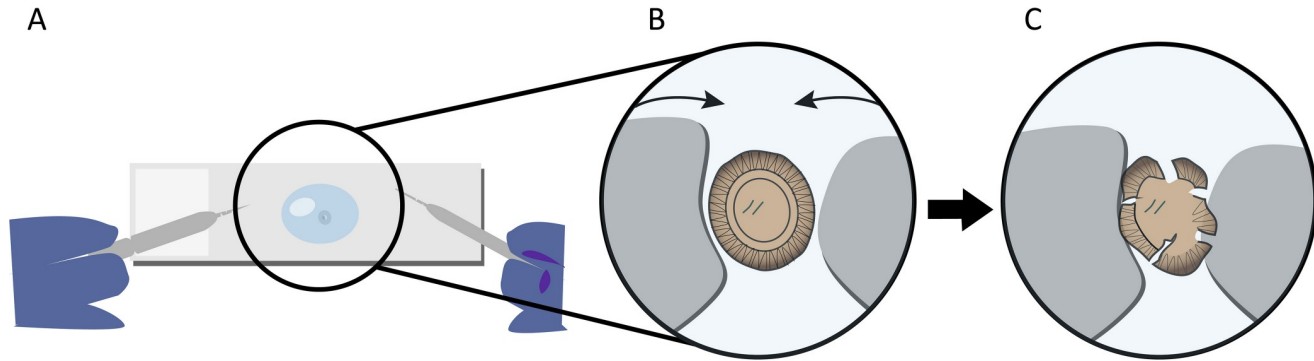

**Fig 2. Disruption of *Taenia* eggs using self-made borosilicate needles. (A)** Place a 4 μL droplet of RNAse-free PBS on a fresh cover slide and add two *Taenia* eggs, **(B)** place the needles in the droplet and gently move them towards the egg, **(C)** then push the needles against each other with mild force until you can visually confirm the disruption of the egg.

## Custom single-cell RNA-sequencing of the samples

Upon arrival at the sequencing facility, sample quality was assessed using an Agilent 2100 Bioanalyzer and samples with sufficient RNA quality were used for library construction using the non-stranded SMART-Seq 2 protocol [24]. Sequencing was done on the BGISEQ-500 platform [25]. We modified this protocol by employing a modified SMART-Seq buffer containing a final concentration of 18.5 mM dATP, 18.5 mM dCTP, 18.5 mM dGTP, 18.5 mM dTTP, 7.4 U/μL RNAse Inhibitor, and 0.7% Triton-X100 and by proceeding immediately to Whole Transcriptome Amplification without carrying out the lysis step published in the original SMART-Seq 2 protocol. The remaining steps of the library preparation are based on the SMART-Seq 2 protocol with the reverse transcription of poly(A)$^{+}$ RNA and template switching being carried out using oligo(dT) primers containing template-switching oligos (TSOs) and cDNA being amplified using PCR with indexed primers. Circularization and sequencing were performed based on the DNA nanoball technology proprietary to BGI.

## Determination of the full-length *Taenia solium* mitochondrial DNA sequence

Donors of *Taenia* ssp. positive stool samples were treated according to the WHO recommendations with an antihelminthic drug (2 g niclosamide) and received a PEG-based purgative (100 g Movicol in 2 liters of water). Stool was collected for 24 hours in a bucket, sieved, and proglottids were collected from the sieve. No scolex was observed. Whole genomic DNA was isolated from *T. solium* proglottids after mechanical disruption with a Potter-Elvehjem tissue homogenizer, followed by proteinase K digestion and DNA-cleanup using the Qiagen Powersoil Pro Kit on a QiaCube Connect device according to the manufacturer's protocol (Qiagen, Hilden, Germany). The presence of *T. solium* DNA was first confirmed using the established nested PCR-assay [13], then 10 ng of DNA were used for preparation of a shotgun sequencing library using the QiaSeq FX kit according to the manufacturer's protocol (Qiagen, Hilden, Germany). The shotgun library was sequenced on a MiSeq next generation sequencer (Illumina) in a 2 x 250 bp paired-end sequencing run generating a total of 5 Mio reads, which were aligned to the 122,393,951 bp *size T. solium* genome (GenBank PRJNA170813) and the 13,709 bp size *T. solium* mtDNA (Genbank AB086256.1) using the BWA MEM v0.7.12 program on a Linux workstation [26]. In order to specifically tweak out the specific mtDNA base pair positions of the here investigated *T. solium* from Eastern Zambia, we performed a secondary local realignment of all fragments that had originally aligned to the AB086256.1 sequence. 67,772 fragments of 250 bp length were thus locally realigned using the Alfred v0.2.3 software [27] generating the full length *T. solium* mtDNA sequence that was placed in GenBank under the accession number MW718881 (https://www.ncbi.nlm.nih.gov/nuccore/MW718881, **S1 Fig**). This sequence was clustal aligned with other available full and partial *T. solium* mtDNA genomes (MW718881; AB086256.1/NC_004022.1; KT591612.1) from Zambia, China, and Peru, respectively, using the Clustal omega v1.2.4 program [28]. The clustal alignment is shown on **S2 Fig** and **S1 Alignment.**

## Bioinformatic processing of sequencing data

Single end FASTQ files were downloaded *via* FTP from the sequencing facility site and quality checked using FastQC [29]. Sequences with PHRED scores below 20 were removed. Single egg data were then gap-aligned to the whole genomes of *T. solium* (NCBI BioProject PRJNA170813 [30]), *T. saginata* (NCBI BioProject PRJNA71493), and *T. asiatica* (NCBI BioProject PRJNA299871) using the STAR v2.7.5 aligner with a stringent setting [31]. Alignment

resulted in an indexed *.bam file that could be inspected using IGV viewer 2.3 [32]. Because the mtDNA does not contain introns, we realigned the single-egg FASTQ-files *via* ungapped single-end alignment to the mtDNA sequences of *T. solium* (GenBank MW718881), *T. saginata* (GenBank NC_009938.1), and *T. asiatica* (Genbank NC_004826.2) using BWA MEM v0.7.12. Later we reduced the paired-end reads for alignment by selecting the first 5 Mio reads from each FASTQ file and reran the alignment program comparing the CPU-time of our CentOS7 Server (running 10 cores) with a single Intel CORE i7 7th Generation processor on a standard laptop personal computer. The details of the alignments and the benchmarking can be inspected in **S1 Fig** and **S1** and **S2 Tables**.

## Results

### Estimation of the worldwide prevalence of *Taenia solium* infestation based on published data

The combination of search terms yielded n = 384 original research publications from PubMed. N = 176 of these publications contained figures about the regional burden of *T. solium*. We then performed an export of all geographic information mentioned in the research articles and plotted these locations on a world map, in order to highlight the areas for which local information about *T. solium* was available (yellow dots on **Fig 3**). These results capture the quantity of

Population at risk for *Taenia solium* infection and international research activities

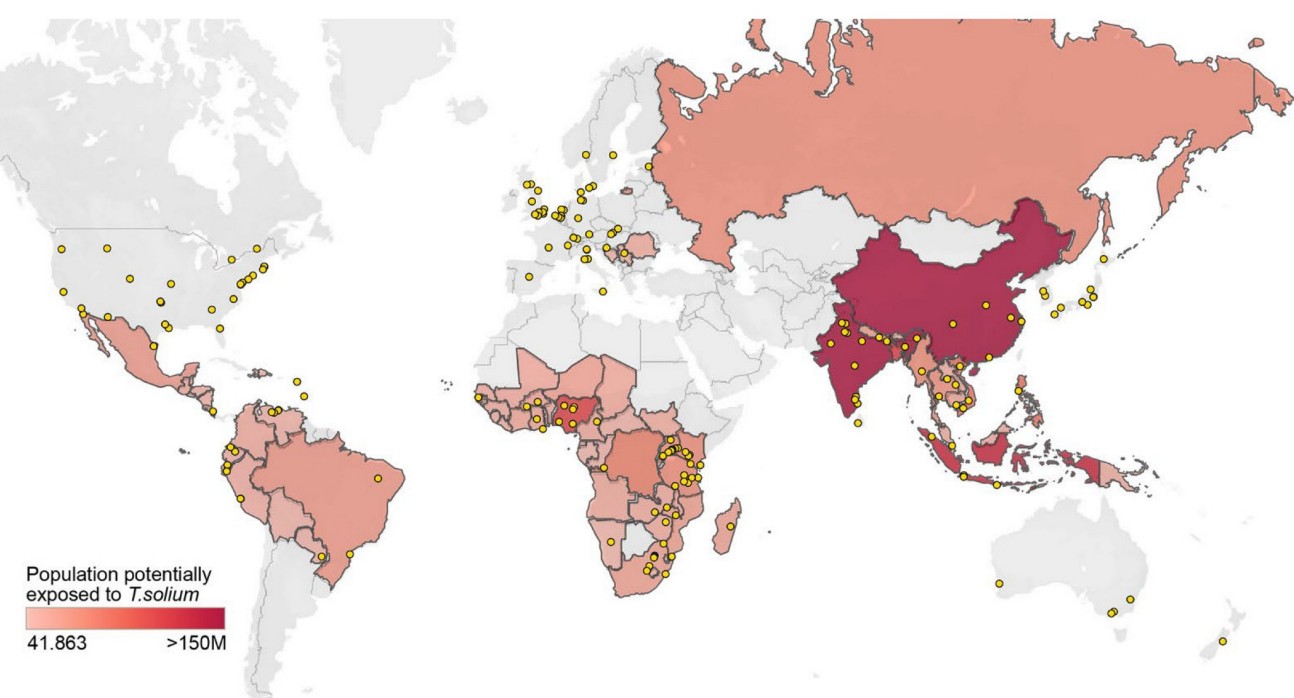

**Fig 3. Mapping of information on *Taenia solium* transmission *versus* size of population with an increased risk for taeniasis.** The color scheme indicates the size of the population exposed to *T. solium*, with dark red indicating a large population at risk for infection. The color code has been generated through a combination of epidemiology data from the WHO and the indices for living conditions provided by the World Bank. Each yellow dot represents a location mentioned in a publication about *T. solium* prevalence. Sources: Base map and data were obtained from OpenStreetMap and OpenStreetMap Foundation. The map contains information from OpenStreetMap and OpenStreetMap Foundation, which is made available under the Open Database License (https://www.openstreetmap.org/#map=2/20.5/22.0).

scientific information that is available to inform health programs or could be used for modeling approaches to better distribute health resources in affected areas.

We investigated how the density of such information would overlap with regions where the population is at high risk to be exposed to *T. solium*. We achieved that by combining WHO and World Bank data and applying a color scheme to each country. Dark red indicates countries with a large population at risk for *T. solium* exposure and infection (**Fig 3**). Many of these countries, however, only show a low density of investigations that pertain to the regional transmission of *T. solium*.

## Sample stability

To develop a better understanding of the robustness and stability of RNA inside *Taenia* eggs from native, unfrozen stool samples, we recorded transportation times as well as storage conditions of the study samples (**Fig 4A**). We then combined the RNA quality data obtained on an Agilent 2100 Bioanalyzer with this transport information and found that large proportions of the *Taenia* RNA inside the eggs remained intact even after storage at 4˚C for five days, transportation for three days at 4–20˚C and a single freeze-thaw cycle at -80˚C = > 4˚C. (**Fig 4B**).

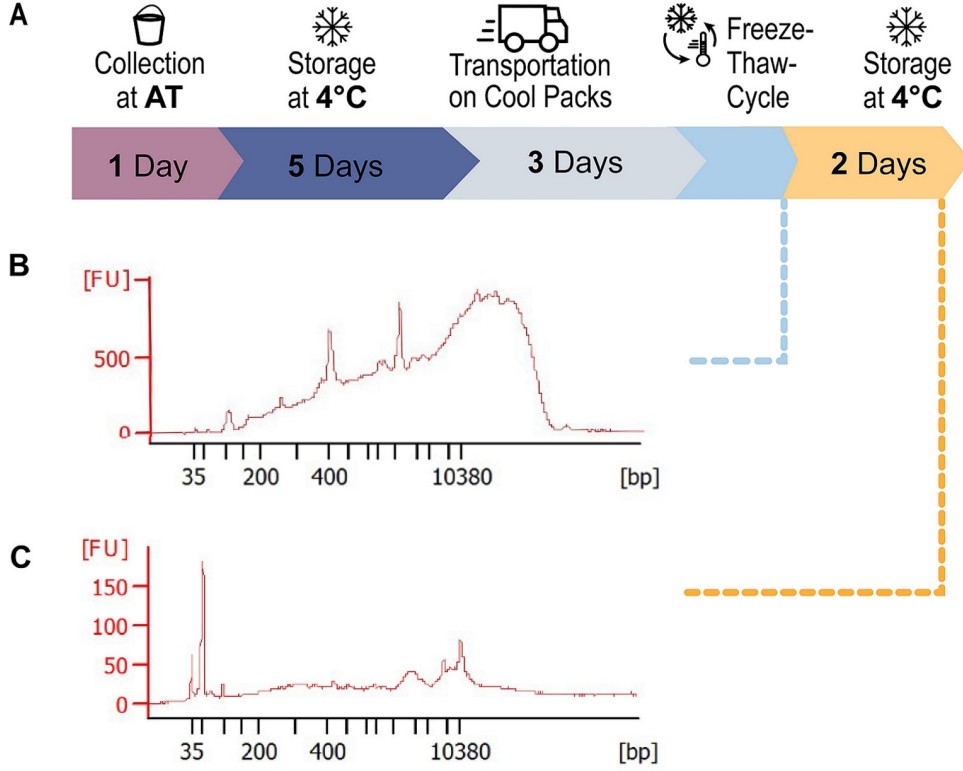

**Fig 4. Impact of pre-analytic storage and transport conditions on RNA quality. (A)** Our experimental storage and transport conditions comprised the following phases: collection at ambient temperature in Zambia, storage at 4˚C for 5 days (**without freezing!**), subsequent transport on coolpacks, and final deep freezing at -80˚C. **(B)** RNA-quality assessed on an Agilent 2100 Bioanalyzer after samples had been prepared using our workflow immediately after thawing before library preparation. Fluorescence Units (FU) as a value of RNA abundance are plotted against the size of the RNA molecules. The size of the RNA-fragments in [bp] is given on the X-axis. The high amount of long RNA molecules and the presence of 18S and 28S rRNA peaks indicates sufficiently good RNA integrity. **(C)** RNA-quality of visually intact eggs, that were kept for 2 days at 4˚C after thawing from -80˚C. The overall low FU-values and the high amount of very short RNA fragments indicates that RNA-quality rapidly deteriorates after thawing of the eggs. Therefore, we advise against several freeze-thaw cycles or longer storage after thawing.

In contrast, we found that RNA was rapidly degraded during storage at 4°C for 48 hours after the eggs had gone through a freeze-thaw cycle (**Fig 4C**). The good RNA quality of samples #1 and #2 allowed us to further analyze the eggs using RNA-sequencing.

## Disruption of a low number of *Taenia* spp. eggs under visual control

Attempts to disrupt *Taenia* spp. eggs by incubation in TRIzol for 48 hours at room temperature with subsequent freeze-thaw cycles either in TRIzol or water were unsuccessful, as was the attempt to disrupt the eggs (n = 8) by a bead-mill approach. Therefore, we opted for a simple disruption protocol under direct visual control, which can be done in a clean, RNAse-free environment with immediate freezing of the sample once disruption has been achieved.

## Shipment and transportation

Next, we established a shipping process that would allow unbureaucratic international transportation of the samples prepared by this protocol. Due to the visually confirmed destruction of all *Taenia* eggs, samples can be shipped as non-infectious lysates containing protein and nucleic acids. The SMART-Seq buffer does not contain hazardous substances, further simplifying sample logistics. As a result of our attempts to recycle the dry ice from the delivery of the SMART-Seq buffer, we show that a small Styrofoam box can keep samples refrigerated on dry ice for up to 4 days under ambient temperature. If leftover dry ice is not available in sufficient quantity, dry ice can be easily created from widely available compressed $CO_2$-bottles [33]. We provide packing, declaration, and labeling instructions that can be found on the protocols.io protocol and allow easy adoption of the "mail-order sequencing" workflow.

## Establishment of the African *Taenia solium* mtDNA sequence

Low-coverage shotgun sequencing of DNA isolated from *T. solium* proglottids and eggs from Zambia on a MiSeq Next Generation Sequencer (Illumina) yielded a total of 4 million reads. Ungapped BWA MEM v0.7.12 alignment covered the *T. solium* genome on average 8.45 fold (**S2 Table**). In contrast, the *T. solium* mtDNA (AB086256.1) was covered on average 1,033.25 fold, thereby securing an unbroken line of coverage over the entire mtDNA molecule (**S2 Fig and S1 Alignment**). This allowed the assembly of the full mtDNA sequence of the *T. solium* species from Zambia, which has now been deposited in GenBank under the accession number MW718881. This reference sequence has been used for further downstream analysis of the mtDNA RNA sequences of the *T. solium* mtDNA transcriptome. The raw FASTQ files from the *T. solium* genome sequencing and the *.bam file from the MW718881 mtDNA alignment can be accessed through the GEO database under the accession number GSE175668 (https://www.ncbi.nlm.nih.gov/geo/query/acc.cgi?acc=GSE175668)

## Highly abundant targets in the *Taenia solium* transcriptome

The samples were analyzed *via* mail-order sequencing using the non-stranded SMART-Seq 2 protocol on the BGISEQ-500 platform. RNA sequencing yielded for both biological replicates a total of 121 million 100 bp paired-end reads per sample (**S1 Table**). To identify suitable targets for diagnostic sequencing, we ranked the transcripts according to their coverage and found that the by far highest ranking transcripts mapped to the mitochondrial genome (**Fig 5**). The average FPKM values of mtDNA derived RNAs from samples #1 and #2 (EN1dxA, EN2dxA) were between 1,700–2,500, while nuclear genes did only produce FPKM values averaging between 52–57 (**S2 Table**). In the next step, we calculated FPKM values for highly abundant mitochondrial genes and compared them to highly expressed housekeeping genes.

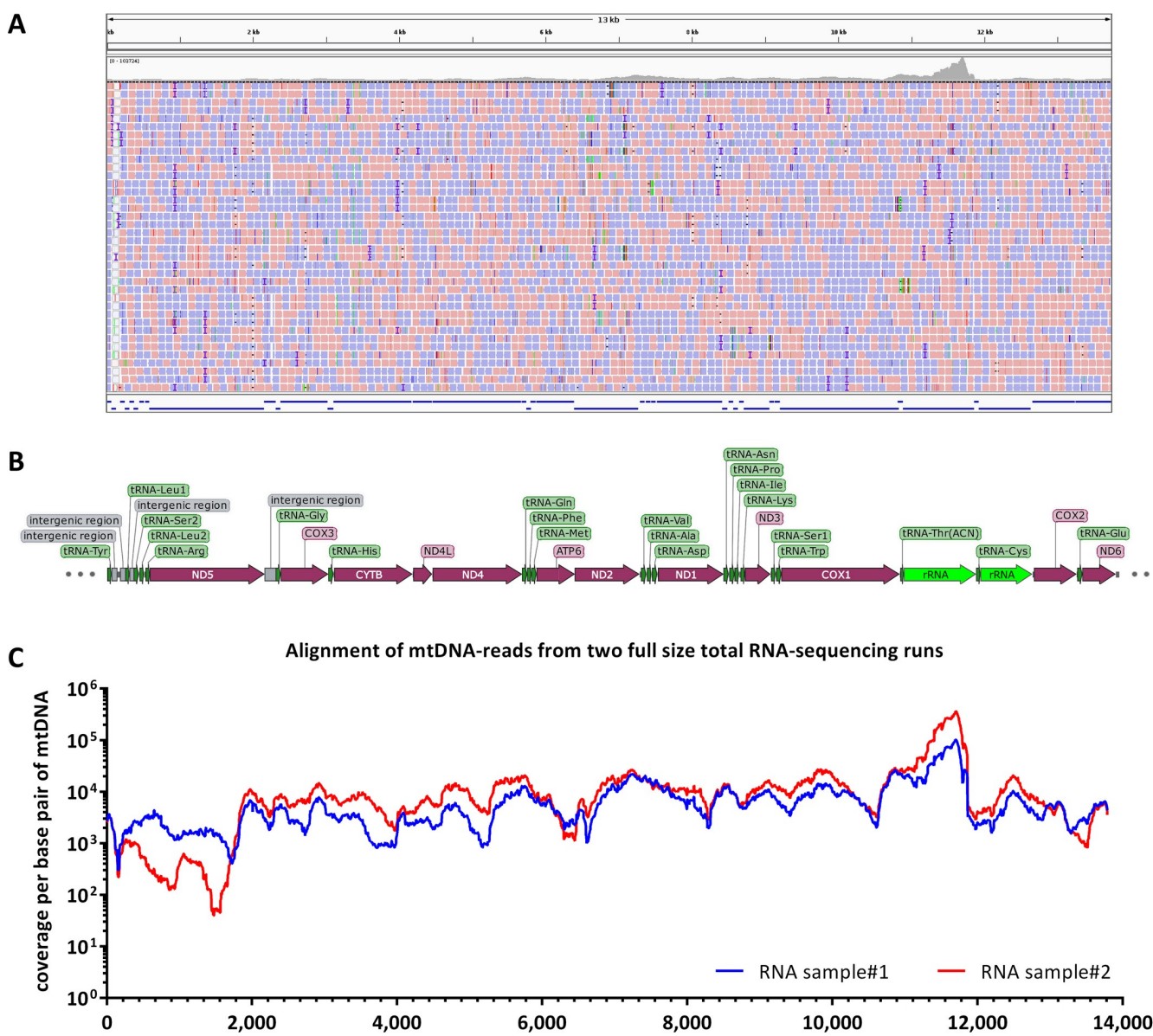

**Fig 5. BWA MEM v0.7.12-alignment and per base coverage of the *Taenia solium* mitochondrial DNA (mtDNA).** (A) Screenshot of the ungapped alignment visualized with the IGV v2.3 viewer. Pink reads represent the forward, blue reads, the reverse sequence reads. (**B**) Annotation of the different genes on the *T. solium* mtDNA. All three panels A-C are drawn to the same scale and correspond to each other. (**C**) Per base coverage of the different transcripts from the mitochondrial genome. The coverage fluctuates between 4 orders of magnitude with highest coverages found for the mitochondrial ribosomal RNA and lowest for the ND5-subunit of mitochondrial complex I. Both samples #1 and #2 show the identical fluctuations across the mtDNA sequence.

Within the mtDNA, the mitochondrial ribosomal RNA sequences had the highest FPKM values (**Fig 6**). The table with all FPKM values of genes expressed in *T. solium* eggs can again be accessed through GSE175668.

We are aware of the fact that FPKM values are not always well suited to compare mRNA expression levels between samples [34]. However, here we compared the expression levels between genes of the same sample in two biological replicates from the same species and tissue that had been processed in parallel, regarding RNA-extraction, poly(A)$^+$-selection, and NGS-

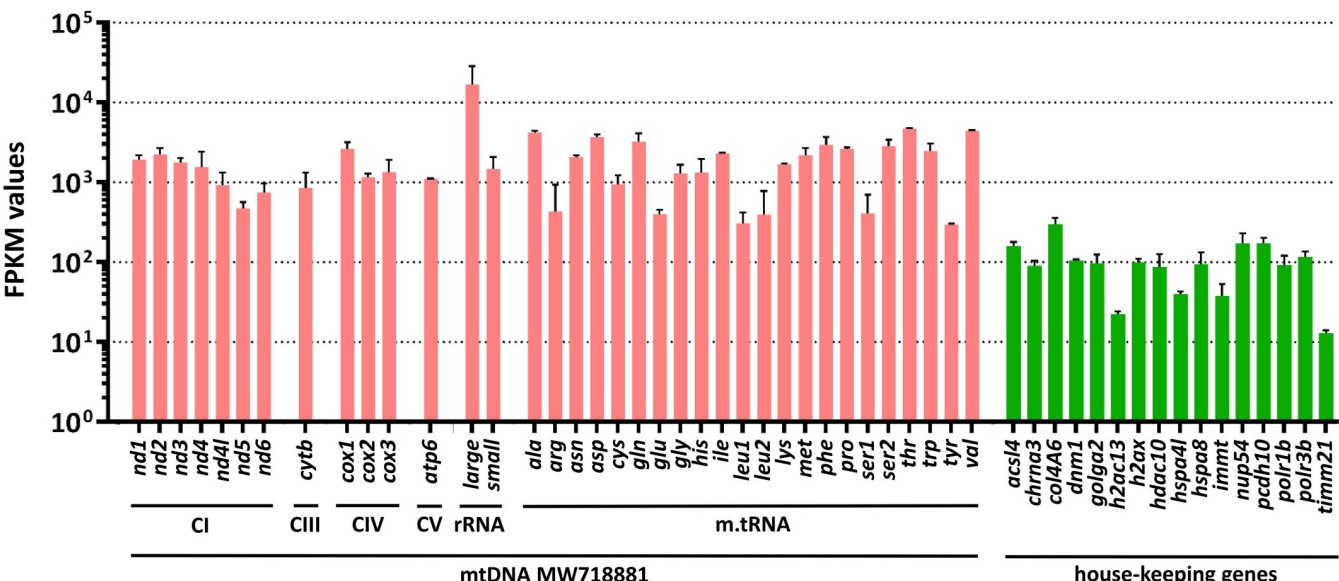

**Fig 6. mRNA gene expression in *Taenia solium* eggs.** The bars depict the FPKM values (fragments per kilobase of exon per million reads mapped) of mitochondrial (mtRNA, pink) and nuclear (nRNA, green) encoded house-keeping genes in single *T. solium* eggs. The expression levels of worm mtDNA encoded genes are on average 10–100 fold higher than that of nuclear encoded house-keeping genes. The mtDNA encoded genes are sorted according to their coding function for subunits of respiratory chain complex I (CI), complex III (CIII), cytochrome C oxidase (CIV), ATPase (CV), the large and small fragments of the mitochondrial ribosomal RNA (rRNA), and the mitochondrial transfer-RNAs (m.tRNA). The bars depict the mean and SD of two biological replicates. ***acsl4***, acyl-CoA synthetase long chain family member 4 (TsM_000719800); ***chrna3***, cholinergic receptor nicotinic alpha 3 subunit (TsM_000436900); ***col4a6***, collagen type IV alpha 6 chain (TsM_000925000); ***dmn1***, dynamin 1 (TsM_000700600); ***golga2***, golgin A2 (TsM_000292400); ***h2ac13***, H2A clustered histone 13 (TsM_000346300); ***h2ax***, H2A.X variant histone (TsM_000426400); ***hdac10***, histone deacetylase 10 (TsM_000886700); ***hspa4l***, heat shock protein family A (Hsp70) member 4 like (TsM_000256500); ***hspa8***, Heat shock protein family A (Hsp70) member 8 (TsM_000375700); ***immt***, Inner membrane mitochondrial protein (TsM_000168900); ***nup54***, Nucleoporin 54 (TsM_000540600); ***pcdh10***, Protocadherin 10 (TsM_000478100); ***polr1b***, RNA polymerase I subunit B (TsM_000754600); ***polr3b***, RNA polymerase III subunit B (TsM_000691900); ***timm21***, Translocase of inner mitochondrial membrane 21 (TsM_001020500), the accession numbers in brackets refer to the transcripts from the PRJNA170813 BioProject [30]. *Nota bene*, due to the large expression differences we opted for a logarithmic rendition of FPKM values.

runs. For FPKM normalization with the StringTie algorithm [35] we did not remove highly expressed genes, because it was the aim of this study to exactly identify those highly expressed genes and to provide a measure for their abundance in comparison to "standard" house-keeping genes.

## RNA sequencing distinguishes between *Taenia* species

In order to assess the suitability of mitochondrial transcriptome analysis for distinguishing *T. solium* from other *Taenia* species, which was the original aim of our investigation, we BWA MEM v0.7.12-aligned the single egg sequencing reads to the different mtDNA sequences of *T. solium*, *T. saginata*, and *T. asiatica*. The resulting *.bam files from the alignments were loaded into the IGV viewer to visually compare the matches of the sequencing reads to the mitochondrial genomes of *T. solium*, *Taenia saginata*, and *Taenia asiatica* (**Fig 7**). Only alignment to the *T. solium* mtDNA yielded an unbroken coverage over the entire mtDNA sequence (**Fig 5**). Alignment to *Taenia saginata* and *Taenia asiatica* only yielded local alignments at highly conserved portions of the mitochondrial genome in the areas of the *cox1* gene and the mitochondrial RNA transcripts (**S3 Fig** and **S2 Alignment**), thereby easily highlighting the taxonomic differences (**Fig 7**). The percentages of mapped reads for *T. saginata* and *T. asiatica* mtDNA are below 0.5% of those mapped to *T. solium* mtDNA (**S1 Table**).

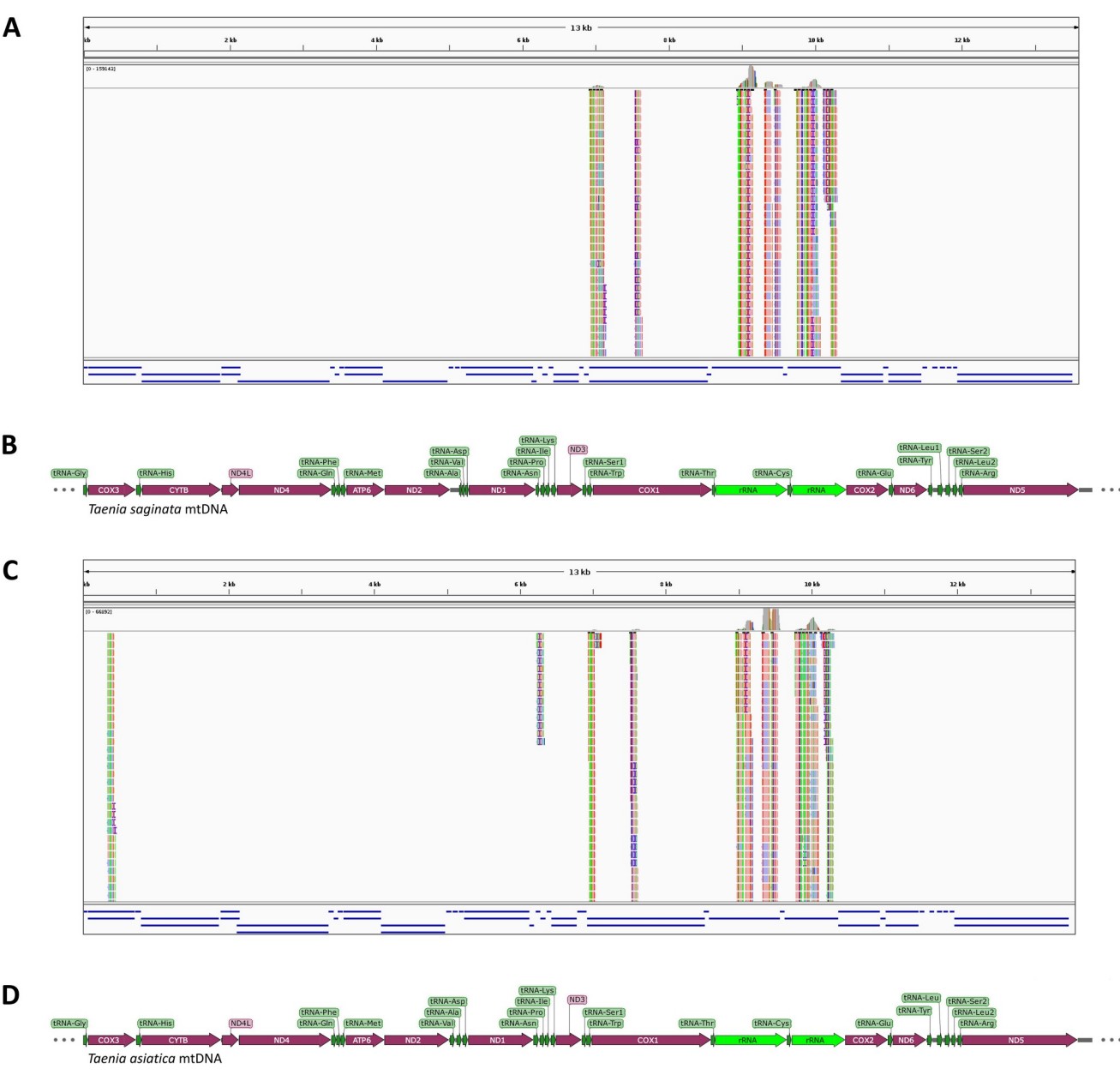

**Fig 7. BWA MEM v0.7.12-alignment of the *Taenia solium* mitochondrial RNA to the mtDNA of other *Taenia* species.** Alignment can only be found in areas of high conservation at the mtDNA genes encoding the subunit I of the cytochrome C oxidase (cox1) and the small and large mitochondrial rRNA fragments. (**A**) IGV visualization of the alignment to the *T. saginata* mtDNA, (**B**) annotation of the *Taenia saginata* mtDNA. (**C**) IGV visualization of the alignment to the *Taenia asiatica* mtDNA, (**D**) annotation of the *Taenia asiatica* mtDNA.

### Bioinformatic workflow for RNA-sequencing on a multicore server *versus* a standard personal computer

The bioinformatic processing of the 121 million RNA-sequencing reads was done on a 64 core Centos7 server with 512 GB RAM, which would not be easily accessible in the field. Hence, we next assessed whether we could further reduce cost by only ordering a 5 million fragment RNA-sequencing depth, knowing that the mtDNA-derived sequences would still be highly abundant to guarantee an unbroken alignment. At the same time, the bioinformatic

calculations using the BWA MEM v0.7.12-software could be run on a standard personal computer with a Linux operating system. We decreased the sequencing density by 24-fold to 5 million RNA sequencing reads, a reduction that would significantly reduce costs and the size of the data set.

For bioinformatic processing we used an average sized notebook with 32 GB RAM and one Intel Core i7 processor. As most of the command line programs use Linux, we installed Ubuntu 14.04 alongside the standard Windows operating system. For sequence alignment we used the same open source programs as on the server (e.g. BWA MEM v0.7.12) with the identical settings as described above. Indeed, the notebook was able to align the reduced dataset of 5 million 100 bp-paired-end reads to the different *Taenia* mitochondrial genomes within around one hour (**S1 Table**). The reduction of sequencing depth from 120 to 5 million still provided an >100-fold average coverage across the entire mitochondrial genome, which is well above the 20x coverage generally accepted for diagnostic purposes and detection of mutations [36]. The resulting *.bam files with alignment to the mitochondrial genomes of *T. solium*, *T. saginata*, and *T. asiatica* were loaded into the IGV viewer for inspection (**Figs 5** and **7**).

## Discussion

Until presently, the prevalence of *T. solium* infections is unknown in large regions across the globe. This makes it difficult to extend targeted preventive health interventions towards regions with a high incidence of *T. solium* infections. To deliver on the WHO goal to control *T. solium* in endemic areas by the year 2030, more data on regional prevalence and transmission are urgently needed [22]. This calls for a simple approach that would enable health personnel to identify taeniasis worldwide [21].

Despite the increasing number of epidemiological investigations on *T. solium*, our mapping and survey of studies highlights the fact that data sources are unevenly distributed across the globe with some regions being well studied, while other countries are covered with only a small number of studies in relation to their population size. The low data density hampers the design of effective disease control programs. This calls for a whole community effort involving all levels of diagnostic laboratories in order to rapidly increase the density of epidemiological data worldwide. This approach requires a simple affordable diagnostic test with high specificity and practicability under different local conditions. With "mail order" RNA-sequencing we provide such a diagnostic method. The community approach we have in mind could be paralleled to the Wikipedia project, where thousands of lay people have jointly created the largest ever encyclopedia [37]. Such locally generated data could then be fed into a central notification system that would extend well beyond small-scale research driven screening programs and would require a central freely accessible database informing the health authorities about potential hot spots for TSCT.

Here we describe an RNA-based next generation sequencing method and workflow that identifies *T. solium* taeniasis reliably, even in low resource settings by staff without molecular diagnostic expertise. The entire workflow is based on a few *Taenia* ssp. eggs extracted from a diagnostic stool sample that might have been transported over large distances on coolpacks under tropical conditions. Currently it is impossible to routinely diagnose *T. solium* taeniasis based on such samples with the microscope alone, because *T. solium* eggs cannot be visually distinguished from the eggs of other tapeworm species like *T. saginata* or *T. asiatica*.

Having the capability to establish the exact diagnosis of a *T. solium* infection has important ramifications on two levels: **(i)** on individual patients, and **(ii)** on public health. On the **individual level**, the choice of the anthelmintic drug to treat a patient with proven taeniasis, e.g praziquantel (PZQ) or albendazole (ABZ) *versus* niclosamide, depends on the *Taenia* species.

As a certain percentage of patients with *T. solium* taeniasis may also suffer from latent NCC, destruction of the parasites by PRZ or ABZ and the subsequent immune reaction may convert asymptomatic cysticerci into symptomatic ones, thereby triggering brain edema, increased intra-cranial pressure, and cerebral seizures [38,39]. Under these circumstances niclosamide would be the drug of choice for unsupervised treatment, because it only acts in the gut lumen with little systemic resorption. Also on the **public health level**, it would be crucial to determine the exact *Taenia* ssp. in order to alert the health authorities about a public health threat in a given area and to initiate mass treatment with an appropriate antihelminthic drug [40]. If a *Taenia* ssp. egg positive stool sample is identified in a routine diagnostic laboratory and public health workers want to confirm a *T. solium* infection, this would imply a time consuming active backtracking of the sample donors in remote rural villages, the collection of proglottids or scolices over 2–3 days, and a histopathological staining and investigation by a versed pathologist. Our workflow allows health personnel to skip these laborious steps and allow pinpointing areas endemic for *T. solium* more easily by simply plotting the geographic origin of the positive stool samples on a map and treating village communities and their livestock accordingly.

Our workflow contributes to the public health toolbox in low-resource settings. In contrast to better funded focused epidemiological surveys, e.g. in Zambia [23], India [41], Peru [42], Lao PDR [43], and China [44] just to mention a few examples, that rely on dedicated technical infrastructure to perform PCR-reactions or immunoassays to detect taeniasis, our approach can help extend taeniasis diagnostics more broadly. The easy readout of the entire mtDNA sequence by single egg RNA-sequencing provides valuable insight into the regional transmission and spread of *T. solium* infections, because the "barcodes" of mitochondrial genes can be easily used for taxonomic fine tuning [45,46] and thus for elucidation of chains of infection.

We found that intermediate storage and transfer of the native stool samples on coolpacks for up to eight days under tropical conditions and storage at -80˚C still yields RNA preparations that allow the exact *T. solium* genotyping, provided that RNA is isolated immediately after taking the eggs from the -80˚C freezer. Finally, we describe a bioinformatic pipeline based on open access tools that allows analysis of next generation sequencing data on an ordinary notebook computer, thereby circumventing the need for expensive high performance computer clusters.

The main limitation for the success of our workflow is the generally low number of routine diagnostic stool samples coming from regions with low socioeconomic status, making it mandatory for health authorities to additionally monitor infestation by low level epidemiological field screenings [23,47]. A second limitation is the fact that not all infested people excrete *Tania* eggs in their feces, at least not all the time. However, given the vast population likely exposed to *T. solium* and the fact that a single diagnosis of *T. solium* taeniasis can direct health measures and further sample collections to an affected region, our method and workflow offers great potential to improve the overall knowledge on *T. solium* prevalence. A third limitation for widespread up-scaling of our workflow are the sequencing costs. However, we present a target for sequence analysis, e.g. the mitochondrial RNA, that is highly expressed in the *Taenia* ssp. eggs, allowing reduction in RNA-sequencing depth to a mere 5 million reads, or even lower. This still offers an overall 100 fold coverage of the mtDNA genes, which is well above the accepted coverage for diagnostic purposes of 20x, bringing down the sequencing costs to under EUR 90 per sample.

## Conclusion

Our method and workflow offers a highly specific solution that can be employed to identify *T. solium* in stool samples without major investment into equipment or training. With our

approach, we propose the analysis of mitochondrial RNA from tapeworm eggs, which we found to be preferable over nuclear DNA or even mtDNA due to its extremely high copy numbers. The high stability of RNA in *Taenia* ssp. eggs over many days even under tropical conditions [3] facilitates sample collection and transport, and even the presence of a few eggs in a stool sample is sufficient to establish a definite diagnosis. Further, we present the first transcriptome data of *T. solium*, that are deposited in a public repository (GEO database) and can be used by other researchers to verify the bioinformatically predicted genes that are transcribed from the *T. solium* genome [30]. Our workflow enables regional healthcare providers to better identify *T. solium* taeniasis and other *Taenia* ssp. and it may be a valuable tool for prospective epidemiological studies into the prevalence of *T. solium*.

## Supporting information

**S1 Fig. Per base coverage of the BWA MEM v0.7.12 alignment of 5 million paired-end fragments from genomic DNA fragments (green line) extracted from a proglottid (PROG1, S2 Table) and from the reduced dataset of 5 million paired-end reads from the RNA-samples #1 (EN1dxA, blue line) and #2 (EN2dxA, red line).** Please note the even distribution of coverage from the genomic fragments and the highly variable distribution on the mitochondrial RNA level. Nota bene, no inference can be made from this figure regarding the absolute amounts of mtDNA copies versus mtRNA transcripts, because RNA and DNA are not extracted from the same cell(s). For relative mapping quantities, please refer to the average coverage as well as the FPKM values on S2 Table).
(TIF)

**S2 Fig. Partial clustal alignment of three different *Taenia solium* mtDNA genomes of samples from Zambia (MW718881), China (NC_004022.1), and Peru (KT591612.1).**
(TIF)

**S3 Fig. Partial clustal alignment of three different *Taenia* species.** *T. solium* (MW718881), *T. saginata* (NC_009938.1), and *T. asiatica* (NC_004826.2).
(TIF)

**S1 Alignment. Full clustal alignment of three different *Taenia solium* mtDNA genomes of samples from Zambia (MW718881), China (NC_004022.1), and Peru (KT591612.1).** The full alignment can be visualized by opening the S1 Alignment.jvp file with the free software JalView that can be downloaded from http://www.jalview.org/getdown/release/.
(JVP)

**S2 Alignment. Full clustal alignment of three different *Taenia* species.** *T. solium* (MW718881), *T. saginata* (NC_009938.1), and *T. asiatica* (NC_004826.2). The full alignment can be visualized by opening the S2 Alignment.jvp file with the free software JalView that can be downloaded from http://www.jalview.org/getdown/release/.
(JVP)

**S1 Table. Benchmarking of the alignment of the two *Taenia solium* transcriptomes (sample #1, EN1dxA; sample #2, En2dxA) to the whole genome versus mitochondrial genome of *T. solium* and other *Taenia* species.** Alignment of 5 million paired-end FASTQ fragments using a personal computer can be achieved in slightly more than one hour.
(XLSX)

**S2 Table. Results and FPKM values from the mapping runs of genomic and RNA fragments from *T. solium* proglottid (PROG1) and egg-samples (EN1dxA, EN2dxA) using the**

**STAR v2.7.5 (gapped) and the BWA MEM v0.7.12 (ungapped) aligners.**
(XLSX)

## Acknowledgments

We thank Dimitrios Laurin Wagner, Jacob Spinnen, Svenja Nierwetberg, Maria Haschke, Sara Dühnen and the whole iGEM team for their support throughout the iGEM project "diagnost-x" that laid the foundation for this scientific work. We also thank the members of the Sikasunge/Mwape laboratory, Susanne Morales Gonzalez and the members of the Schuelke laboratory for providing the excellent framework to conduct this research.

## Author Contributions

**Conceptualization:** Henrik Sadlowski, Markus Schuelke.

**Data curation:** Henrik Sadlowski, Markus Schuelke.

**Formal analysis:** Henrik Sadlowski, Johannes A. Kuehn, Markus Schuelke.

**Funding acquisition:** Henrik Sadlowski, Andrea S. Winkler, Markus Schuelke.

**Investigation:** Henrik Sadlowski, Jonathan Hiss, Christian G. Schneider.

**Methodology:** Markus Schuelke.

**Project administration:** Veronika Schmidt, Markus Schuelke.

**Resources:** Chummy S. Sikasunge, Andrea S. Winkler, Markus Schuelke.

**Software:** Markus Schuelke.

**Supervision:** Veronika Schmidt, Kabemba E. Mwape, Andrea S. Winkler, Markus Schuelke.

**Validation:** Henrik Sadlowski, Johannes A. Kuehn, Markus Schuelke.

**Visualization:** Henrik Sadlowski, Johannes A. Kuehn, Markus Schuelke.

**Writing – original draft:** Henrik Sadlowski, Markus Schuelke.

**Writing – review & editing:** Henrik Sadlowski, Veronika Schmidt, Jonathan Hiss, Johannes A. Kuehn, Christian G. Schneider, Gideon Zulu, Alex Hachangu, Chummy S. Sikasunge, Kabemba E. Mwape, Andrea S. Winkler, Markus Schuelke.

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
