## [Decision Letter · Decision Letter 0]

1 Nov 2021

Dear Professor Schuelke,

Thank you very much for submitting your manuscript "Diagnosis of Taenia solium infections based on "mail order" RNA-sequencing of single tapeworm egg isolates from stool samples" for consideration at PLOS Neglected Tropical Diseases. As with all papers reviewed by the journal, your manuscript was reviewed by members of the editorial board and by several independent reviewers. The reviewers appreciated the attention to an important topic. Based on the reviews, we are likely to accept this manuscript for publication, providing that you modify the manuscript according to the review recommendations. 

Sincerely,

Xiao-Nong Zhou

Associate Editor

Hélène Carabin

Deputy Editor

Reviewer's Responses to Questions

**Key Review Criteria Required for Acceptance?**

**Methods**

-Are the objectives of the study clearly articulated with a clear testable hypothesis stated?

-Is the study design appropriate to address the stated objectives?

-Is the population clearly described and appropriate for the hypothesis being tested?

-Is the sample size sufficient to ensure adequate power to address the hypothesis being tested?

-Were correct statistical analysis used to support conclusions?

-Are there concerns about ethical or regulatory requirements being met?

Reviewer #1: This is a very clear, well-written, comprehensive and extremely detailed manuscript which satisfies the criteria for publication in PLoS Neglected Tropical Diseases journal. 

1. The objectives are clear, and provided with a testable hypothesis.

2. The population is clearly described and adequate.

3. The sample size is sufficient. 

4. All the analysis is sufficient to support the results and final conclusion.

5. There are no concerns about the ethical or regulatory aspects. 

6. The study has included its limitations with great detail which are very useful for future research. 

7. The flowcharts/diagrams are excellent and deserve appreciation.

Reviewer #2: Yes

**Results**

-Does the analysis presented match the analysis plan?

-Are the results clearly and completely presented?

-Are the figures (Tables, Images) of sufficient quality for clarity?

Reviewer #1: 1. The analysis and results are well explained, detailed and answer the questions provided by the hypothesis.

2. The results are clear and completely presented. The flowcharts/diagrams are excellent and deserve appreciation. 

3. The figures are of very good quality.

Reviewer #2: Yes

**Conclusions**

-Are the conclusions supported by the data presented?

-Are the limitations of analysis clearly described?

-Do the authors discuss how these data can be helpful to advance our understanding of the topic under study?

-Is public health relevance addressed?

Reviewer #1: 1. The conclusions are supported by the data.

2. The limitations are detailed and well explained. 

3. The authors have discussed how the data can be useful for future research.

4. The public health relevance has been addressed.

Reviewer #2: Yes

**Editorial and Data Presentation Modifications?**

Reviewer #1: I recommend the manuscript be accepted. 

Minor edit :- Line 150: "'Here we present...'" (Please insert the word 'we' here).

Reviewer #2: (No Response)

**Summary and General Comments**

Reviewer #1: The manuscript under review focuses on Taeniasis, which is a neglected tropical disease caused by the parasite Taenia solium, and it offers a protocol to identify the presence of the parasite within stool samples, by analyzing the mitochondrial RNA from tapeworm eggs. The workflow combines both traditional and new, advanced techniques which has the potential of assisting researchers in diagnosing the disease easily, based on very few eggs present in the stool sample. The entire manuscript is well written, detailed, easy to understand and comprehensive, and is of relevance in the world of neglected tropical diseases, which often get overlooked and underfunded.

Reviewer #2: I reviewed the manuscript (PNTD-D-21-01324) entitled "Diagnosis of Taenia solium infections based on "mail order" RNA-sequencing of single tapeworm egg isolates from stool samples". Here attached my some comments below.

Major Comments:

In this MS, the authors developed a novel method called “mail order” single egg RNA-sequencing, which can precise identify the exact Taenia ssp. just by using a few eggs found in routine diagnostic stool samples. Meanwhile, the authors provided the first transcriptome data of T. solium, which can be used by other researchers to verify the bioinformatically predicted genes that are transcribed from the T. solium genome. Although with small limitations, the workflow developed in this study enables regional healthcare providers to better identify T. solium taeniasis and other Taenia ssp. and it may be a valuable tool for prospective epidemiological studies into the prevalence of T. solium. Summarily, the work is interesting and valuable, I recommend publication.

Specific Comments:

Line 192, “for up to five days”, I want to know how long the samples can be stored at most?

Line 218, “Preparation of borosilicate needles for egg disruption”, the process is a little complex, is there any substitute that can by purchased?

Line 238, “8 Taenia eggs”, why select 8? fewer than 8 is OK?

Line 261, “Custom single-cell RNA-sequencing of the samples”, as one of the main technology of the developed workflow, I think the authors should pay more detailed descriptions of the custom single-cell RNA-sequencing of Taenia ssp. eggs. 

Line 388, “average FPKM values”, The use of FPKMs for calculating differential expression of genes across samples. This approach has been proven to be unacceptable for the purpose of differential expression analyses. See these references for clarification and alternative methods: "Misuse of RPKM or TPM normalization when comparing across samples and sequencing protocols" - https://rnajournal.cshlp.org/content/early/2020/04/13/rna.074922.120"A survey of best practices for RNA-seq data analysis" - https://www.ncbi.nlm.nih.gov/pmc/articles/ PMC4728800/

PLOS authors have the option to publish the peer review history of their article (what does this mean?). If published, this will include your full peer review and any attached files.

Reviewer #1: No

Reviewer #2: No

Figure Files:

Data Requirements:

Reproducibility:

References

---

## [Editor Report · Decision Letter 1]

22 Nov 2021

Dear Professor Schuelke,

We are pleased to inform you that your manuscript 'Diagnosis of Taenia solium infections based on "mail order" RNA-sequencing of single tapeworm egg isolates from stool samples' has been provisionally accepted for publication in PLOS Neglected Tropical Diseases.

Best regards,

Xiao-Nong Zhou

Associate Editor

Hélène Carabin

Deputy Editor

---

## [Editor Report · Acceptance letter]

5 Dec 2021

Dear Professor Schuelke,

We are delighted to inform you that your manuscript, "Diagnosis of </i>Taenia solium</i>  infections based on "mail order" RNA-sequencing of single tapeworm egg isolates from stool samples," has been formally accepted for publication in PLOS Neglected Tropical Diseases.

Best regards,

Shaden Kamhawi

co-Editor-in-Chief

Paul Brindley

co-Editor-in-Chief
